# An Optimal Geometry Configuration Algorithm of Hybrid Semi-Passive Location System Based on Mayfly Optimization Algorithm

**DOI:** 10.3390/s21227484

**Published:** 2021-11-11

**Authors:** Aihua Hu, Zhongliang Deng, Hui Yang, Yao Zhang, Yuhui Gao, Di Zhao

**Affiliations:** 1School of Electronic Engineering, Beijing University of Posts and Communications, Beijing 100876, China; dengzhl@bupt.edu.cn (Z.D.); yaoo@bupt.edu.cn (Y.Z.); gaoyuhui@bupt.edu.cn (Y.G.); zhaodi820@bupt.edu.cn (D.Z.); 2School of Information Technology, Hebei University of Economics and Business, Shijiazhuang 050061, China; 3Astronaut Research and Training Center, Beijing 100094, China; yangwei1985.com@163.com

**Keywords:** optimal geometry configuration, semi-passive location, GDOP, MOA, TDOA&AOA, UAV

## Abstract

In view of the demand of location awareness in a special complex environment, for an unmanned aerial vehicle (UAV) airborne multi base-station semi-passive positioning system, the hybrid positioning solutions and optimized site layout in the positioning system can effectively improve the positioning accuracy for a specific region. In this paper, the geometric dilution of precision (GDOP) formula of a time difference of arrival (TDOA) and angles of arrival (AOA) hybrid location algorithm is deduced. Mayfly optimization algorithm (MOA) which is a new swarm intelligence optimization algorithm is introduced, and a method to find the optimal station of the UAV airborne multiple base station’s semi-passive positioning system using MOA is proposed. The simulation and analysis of the optimization of the different number of base stations, compared with other station layout methods, such as particle swarm optimization (PSO), genetic algorithm (GA), and artificial bee colony (ABC) algorithm. MOA is less likely to fall into local optimum, and the error of regional target positioning is reduced. By simulating the deployment of four base stations and five base stations in various situations, MOA can achieve a better deployment effect. The dynamic station configuration capability of the multi-station semi-passive positioning system has been improved with the UAV.

## 1. Introduction

At present, people are increasingly dependent on location services. In addition to indoor and outdoor applications, location awareness applications for extreme environments are also increasing, such as searching and rescuing after natural disasters, submarine sensor network positioning, etc. In this paper, the mobile phone positioning system working in an earthquake disaster environment is studied. Because of the complexity of the ruined environment, the positioning accuracy is greatly influenced by the multipath and non-line-of-sight (NLOS) propagation. To increase the positioning accuracy, finding an optimal geometric configuration algorithm for the location system is one of the auxiliary means. Recently, unmanned aerial vehicles (UAV) have been shown to be quite effective in situations such as surveillance and search and rescue (SAR) [1]. This paper mainly studies the SAR problem of a combination of multiple UAVs. In natural disasters, especially earthquakes, rescuing survivors in the shortest possible time is critically important. Using UAVs with location base-stations can help reducing search time because UAVs can provide the location of victims with a telephone in need in an unprecedented and efficient way [2]. The airborne base station is used in the wireless positioning system, which improves the flexibility of the base station layout.

The optimal geometric configuration of the location system can be obtained by different positioning measurement methods. Various types of measurement have been applied for accurate localization including angle of arrival (AOA) [3], time of arrival (TOA) [4], time difference of arrival (TDOA) [5], frequency difference of arrival (FDOA) [6] or received signal strength (RSS) [7] measurements. Bishop, A.N., et al. Studied optimal deployment of homogeneous sensors including TDOA, AOA, and RSS measurement, respectively. The single measurement method is usually used for optimal station layout in 2-D. A unified way to analytically characterize the optimal placements of bearing, range or RSS sensors was proposed in 2D and 3D [8]. The above methods are suitable for indoor or outdoor environments. The single positioning algorithm has low positioning accuracy and is not suitable for application in the complex environment of ruins.

In recent years, there have been some hybrid location algorithms [9,10,11]. These hybrid location algorithms increased the positioning accuracy. Based on the hybrid positioning, some optimal station placement methods have appeared. Wang, et al. proposed the optimal deployment of sensor–emitter geometries for hybrid localization using TDOA and AOA measurements [12]. Shi, et al. proposed a simple solution to the optimal deployment of cooperative nodes in two-dimensional TOA-based and AOA-based localization systems [13]. Wang, et al. proposed optimal sensor deployment with hybrid TDOA and FDOA measurements [14]. Most studies focused on the optimal deployment of homogeneous sensors, i.e., range, bearing and RSS sensors. The above algorithms are mostly used in the layout of indoor wireless sensor networks. So far, no one has studied the optimization of the base station layout of the wireless positioning system in the ruin environment.

The optimal criterion is often used as the objective function of optimization algorithms. The determinant of the Fisher information matrix (FIM) is applied to obtain the optimal criterion [15,16,17,18]. The maximum of determinants (FIM) represents the highest positioning accuracy [19,20]. The other popular index in navigation and positioning systems to measure the configuration is the geometric dilution of precision (GDOP) [13,21]. GDOP is the square root of the trace of Cramer Rao lower bound (CRLB) [22,23], which is the inverse of FIM. The minimum GDOP can also represents the highest positioning accuracy. The constrained optimization algorithm can be used. Wang. applied the interior point penalty function method in both general and co-located scenarios [12]. Some swarm intelligence optimization algorithm, such as PSO, GA [24,25,26,27,28], can be used. Li. studied an optimal geometric configuration for TDOA location systems with Reinforcement Learning [29]. In this paper, we propose an optimal geometry configuration algorithm of hybrid location systems based on MOA, which is for the UAV airborne multi-base station semi-passive positioning system.

In this paper, the location of a ruined environment is complex, and the hybrid location algorithm of TDOA and AOA deduced the target location. The GDOP of the hybrid location algorithm is minimized as the optimization criterion, and GDOP is taken as the objective function. Another swarm intelligence optimization algorithm, the Mayfly optimization algorithm [30], is used to find the optimal layout of the location system.

The remainder of this paper is organized as follows: Section 2 gives the formulae of TDOA and AOA hybrid positioning algorithm obtained from one transmitter and *n* receivers. In Section 3, the Fisher information matrix is derived, then the CRLB and GDOP are obtained. Section 4 presents the solution to the optimal deployment of the location system which is achieved by using swarm intelligence optimization algorithm, and the mayfly algorithm is emphasized. In Section 5, the simulation results illustrate the effectiveness of the proposed approach. Finally, the conclusion of our work is presented and the future research is recommended in Section 6.

## 2. Time Difference of Arrival (TDOA) and Angle of Arrival (AOA) Hybrid Positioning Algorithm

A semi-passive positioning system in the ruin environment is composed of one master base station and *n* slave base stations as shown in Figure 1. The master base station as the transmitter induces the target to send the access request signal (PCH, paging channel), then the *n* base stations capture the access signal (PRACH, physical random access channel), to measure and deduce the TDOA and AOA formulas of the *n* + 1 base stations. Consider a three-dimensional (3D) scenario, the coordinate vector of the master station is expressed as $s_{0}={\left[x_{0},y_{0},z_{0}\right]}^{T}$, the coordinate vector of the slave stations is expressed as $s_{i}={\left[x_{i},y_{i},z_{i}\right]}^{T},i=1,2,\cdots ,N$ the position coordinate vector of the tested terminal (target) is expressed as $s_{t}={\left[x_{t},y_{t},z_{t}\right]}^{T}$. $d_{i}$ indicates the distance from the positioning target to the base station [31].

### 2.1. TDOA Equations

Here, the TDOA measurement values of different receivers are multiplied by the signal propagation speed to get the distance difference between the target and the corresponding receiver. The distance value of each pair of transceivers is equal to the line-of-sight distance of the corresponding transmitter and receiver plus the TDOA measured by the receiver. There is a linear relationship between TDOA and the distance between the transmitter and receiver. As shown in the figure above, the master station is used as the transmitter, and the distance from the receiver *i* can be calculated by the following formula:(1)$$d_{0,i}=d_{0}-d_{i}=c∆t_{i}$$
where $d_{0,i}$ is the distance between the slave station and the master station, $d_{0}$ is the distance between the target and the master station and $d_{i}$ is the distance between the target and the receiver (the slave station). c is the speed of light, $∆t_{i}$ is the time difference of arrival. The calculation formula is as follows:(2)$$d_{0}=\left\|s_{0}-s_{t}\right\|$$
(3)$$d_{i}=\left\|s_{i}-s_{t}\right\|$$
where ‖·‖ is the $l_{2}$-norm.

Equation (1) can be rewritten as:(4)$$d_{0,i}+d_{i}=d_{0}$$

Submit Equations (2) and (3) into Equation (4), and then squaring the two sides of the equation lead to Equation (5)
(5)$$(s_{i}^{T}-s_{0}^{T})s_{t}-d_{0,i}d_{i}=\frac{1}{2}\left(d_{{}_{0,i}}^{2}+s_{i}^{T}s_{i}-s_{0}^{T}s_{0}\right)$$

Do the calculation of Equation (5) for all slave base stations, and obtain the matrix form:(6)$$A_{T}x_{1}=b_{T}$$
where,
(7)$$\begin{array}{l} b_{T}={\left[b_{T,1}^{T}\begin{array}{cc} \cdots & b_{T,N}^{T} \end{array}\right]}^{T}\\ \left[b_{T,i}^{T}\right]=\frac{1}{2}\left(d_{0,i}^{2}+s_{i}^{T}s_{i}-s_{0}^{T}s_{0}\right)\\ A_{T}={\left[\begin{array}{ccc} A_{T,1}^{T} & \cdots & A_{T,N}^{T} \end{array}\right]}^{T}\\ A_{T,i}^{}=\left[\begin{array}{cc} s_{i}^{T}-s_{0}^{T} & -e_{i}^{T}⊗d_{0,i} \end{array}\right]\\ x_{1}={\left[\begin{array}{cc} s_{t}^{T} & d^{T} \end{array}\right]}^{T}\\ d={\left[\begin{array}{ccc} d_{1} & \cdots & d_{N} \end{array}\right]}^{T} \end{array}$$
where $e_{i}$ is a unit matrix.

### 2.2. AOA Equations

The AOA method is to measure the direction of arrival of the received signal from the azimuth and elevation angles of the receiver to the target, and the azimuth angle *α_i_* and elevation angle *β_i_* of the *i*th receiver can be defined as:(8)$$\alpha_{i}=\tan^{-1}(\frac{y_{t}-y_{i}}{x_{t}-x_{i}})$$
(9)$$\beta_{i}=\tan^{-1}(\frac{z_{t}-z_{i}}{\left(x_{t}-x_{i}\right)\cos\alpha_{i}+\left(y_{t}-y_{i}\right)\sin\alpha_{i}})$$

Define a vector as the compact form:(10)$$\gamma_{i}=\left[\begin{array}{l} \alpha_{i}\\ \beta_{i} \end{array}\right]\begin{array}{cc} & i=1,2,\cdots ,N \end{array}$$

Using the trigonometric formula, Equations (8) and (9) can be represented as:(11)$$x\sin\alpha_{i}-y\cos\alpha_{i}=x_{i}\sin\alpha_{i}-y_{i}\cos\alpha_{i}$$
(12)$$x\sin\beta_{i}\cos\alpha_{i}-y\sin\beta_{i}\sin\alpha_{i}-z\cos\beta_{i}=x_{i}\sin\beta_{i}\cos\alpha_{i}-y_{i}\sin\beta_{i}\sin\alpha_{i}-z_{i}\cos\beta_{i}$$

Then Equations (11) and (12) can be written in the compact form:(13)$$A_{A,i}x_{1}=b_{A,i}\begin{array}{cc} , & i=1,2,\cdots ,N \end{array}$$
where,
(14)$$\begin{array}{l} b_{A,i}=\left[\begin{array}{l} x_{i}\sin\alpha_{i}-y_{i}\cos\alpha_{i}\\ x_{i}\sin\beta_{i}\cos\alpha_{i}-y_{i}\sin\beta_{i}\sin\alpha_{i}-z_{i}\cos\beta_{i} \end{array}\right]\\ A_{A,i}=\left[\begin{array}{cc} \begin{array}{cc} \begin{array}{cc} \begin{array}{c} \sin\alpha_{i}\\ \sin\beta_{i}\cos\alpha_{i} \end{array} & \begin{array}{c} -\cos\alpha_{i}\\ \sin\beta_{i}\sin\alpha_{i} \end{array} \end{array} & \begin{array}{c} 0\\ -\cos\beta_{i} \end{array} \end{array} & O_{2\times N} \end{array}\right] \end{array}$$

Extending Equation (13) to all receivers, the final AOA equation is expressed as:(15)$$A_{A}x_{1}=b_{A}$$
where
(16)$$\begin{array}{l} A_{A}=\left[\begin{array}{ccc} A_{A,1}^{T} & \cdots & A_{A,N}^{T} \end{array}\right]\\ b_{A}=\left[\begin{array}{ccc} b_{A,1}^{T} & \cdots & b_{A,N}^{T} \end{array}\right] \end{array}$$

### 2.3. TDOA and AOA Hybrid Positioning

Combination Equations (6) and (15) yield the hybrid positioning formulation:(17)$$A_{1}x_{1}=b_{1}$$
where,
(18)$$\begin{array}{l} b_{1}={\left[\begin{array}{l} b_{T}\\ b_{A} \end{array}\right]}_{\left(N+2N\right)\times 1}\\ A_{1}={\left[\begin{array}{l} A_{T}\\ A_{A} \end{array}\right]}_{\left(N+2N\right)\times \left(3+N\right)} \end{array}$$

The above equations do not consider the influence of measurement noise, and the measurement values of each parameter in the actual positioning system inevitably have error noise. It is assumed that the measurement errors in TDOA and AOA are Gaussian random errors with a mean value of 0. The position solution analysis with measurement noise is as follows.

Suppose the distance measurement error in TDOA is represented by ω*_T_*, and the measurement error in AOA is represented by ω*_A_*, and the two are independent of each other. It can be expressed as follows:(19)$$\begin{array}{l} d=\hat{d}-\omega_{T}\\ \gamma =\hat{\gamma }-\omega_{A} \end{array}$$
where $\hat{d}$ and $\hat{\gamma }$ are the measured values,d and γ are the correct values, $d=\left[d_{1}^{T},\cdots ,d_{N}^{T}\right]$ and $\gamma =\left[\gamma_{1}^{T},\cdots ,\gamma_{N}^{T}\right]$ the two vectors contain all noise-free measured values. The distance measurement error vector is expressed as $\omega_{T}={\left[\omega_{T,1}^{T},\cdots \omega_{T,N}^{T}\right]}^{T}$, where $\omega_{T,i}=\omega_{0,i}$; AOA measurement error vector is expressed as $\omega_{A}={\left[\omega_{A,1}^{T},\cdots ,\omega_{A,N}^{T}\right]}^{T}$, where $\omega_{A,i}^{T}={\left[∆\alpha_{i}\;∆\beta_{i}\right]}^{T}$.

## 3. Geometric Dilution of Precision (GDOP) Solution Process

In navigation and positioning systems, GDOP is a parameter to measure the geometric layout [13,21]. GDOP is the square root of the trace of CRLB, which is the inverse of FIM. The minimum GDOP represents the highest positioning accuracy. The solution process of GDOP is given below.

### 3.1. Cramer Rao Lower Bound (CRLB) Solution for Position Estimation

In the optimal case, the estimated value of the unbiased estimation variable can reach CRLB. The solution of CRLB can be obtained by inverting the FIM. The Fisher information matrix of the TDOA and AOA hybrid positioning algorithm is given by:(20)$$I\left(s_{t}\right)=E\left[{\left(\frac{\partial \ln p\left(\hat{q};s_{t}\right)}{\partial s_{t}}\right)}^{T}\left(\frac{\partial \ln p\left(\hat{q};s_{t}\right)}{\partial s_{t}}\right)\right]$$
where $p\left(\hat{q},x_{t}\right)$ denotes the probability density function (PDF) of a parameter $\hat{q}$ for unknown variables $x_{t}$, and $\hat{q}=\left[{\hat{d}}^{T}\;{\hat{\gamma }}^{T}\right]$ is a vector that combines the noisy measured values in Equation (19). In the case of Gaussian observations, with covariance matrix $C_{q}$. Then, invert the Fisher information matrix to obtain CRLB.
(21)$$\begin{array}{ll} CRLB\left(s_{t}\right) & =I^{-1}\left(s_{t}\right)={\left[{\left(\frac{\partial q}{\partial s_{t}}\right)}^{T}C_{q}^{-1}\frac{\partial q}{\partial s_{t}}\right]}^{-1}\\ & ={\left[{\left(\frac{\partial d}{\partial s_{t}}\right)}^{T}C_{\omega T}^{-1}\frac{\partial d}{\partial s_{t}}+\left(\frac{\partial \gamma }{\partial s_{t}}\right)C_{\omega A}^{-1}\frac{\partial \gamma }{\partial s_{t}}\right]}^{-1} \end{array}$$


The first part of Equation (21) is distance measurement, and the second part is AOA measurement. $C_{\omega T}$ and $C_{\omega A}$ denote the error covariance matrix of TDOA and AOA, respectively, expressed as:(22)$$\begin{array}{l} C_{\omega T}=E\left[\omega_{T}\omega_{T}^{T}\right]=diag(C_{\omega T,i})\\ C_{\omega T,i}=E[\omega_{T,i}\omega_{T,i}^{T}]\begin{array}{cc} , & i=1,2\cdots N \end{array} \end{array}$$
(23)$$\begin{array}{l} C_{\omega A}=diag\left(C_{\omega A,i}\right),\;i=1,2\ldots .N\;\\ C_{\omega A,i}=E\left[\omega_{A,i}\omega_{A,i}^{T}\right],\;i=1,2\ldots .N \end{array}$$
The first part can be derived from $d=\left[\begin{array}{ccc} d_{1}^{T} & \cdots & d_{N}^{T} \end{array}\right]$


(24)$$\frac{\partial d}{\partial s_{t}}={\left[\left({\frac{\partial d_{1}}{\partial s_{t}}}^{T}\right)\ldots \left({\frac{\partial d_{N}}{\partial s_{t}}}^{T}\right)\right]}^{T}=H$$
where
(25)$$\frac{\partial d_{i}}{\partial s_{t}}=\left[\frac{{\left(s_{t}-s_{0}\right)}^{T}}{\left\|s_{t}-s_{0}\right\|}-\frac{{\left(s_{t}-s_{i}\right)}^{T}}{\left\|s_{t}-s_{i}\right\|}\right]=H_{i\;},\;i=1\ldots .N$$
2.The second part can be derived from $\gamma =\left[\begin{array}{ccc} \gamma_{1}^{T} & \cdots & \gamma_{N}^{T} \end{array}\right]$



(26)
$$\frac{\partial \gamma }{\partial s_{t}}={\left[{\left(\frac{\partial \gamma_{1}}{\partial s_{t}}\right)}^{T}\ldots {\left(\frac{\partial \gamma_{N}}{\partial s_{t}}\right)}^{T}\right]}^{T}=Y$$



Write Equations (8) and (9) in vector form, which can be expressed as:(27)$$\gamma_{i}=[\begin{array}{c} \alpha_{i}=\tan^{-1}(\frac{y_{t}-y_{i}}{x_{t}-x_{i}})\\ \tan^{-1}(\frac{z_{t}-z_{i}}{(x_{t}-x_{i})\cos\alpha_{i}+(y_{t}-y_{i})\sin\alpha_{i}}) \end{array}]$$

Perform partial differential operations on both sides of Equation (25) to obtain as
(28)$$\frac{\partial \gamma_{i}}{\partial s_{t}}=\frac{1}{d_{i}}\left[\begin{array}{ccc} -\frac{\sin\alpha_{i}}{\cos\beta_{i}} & \frac{\cos\alpha_{i}}{\cos\beta_{i}} & 0\\ -\cos\alpha_{i}\sin\beta_{i} & -\sin\alpha_{i}\cos\beta_{i} & \cos\beta_{i} \end{array}\right]=Y_{i}$$


Then CRLB abbreviated form can be obtained as
(29)$$CRLB\left(s_{t}\right)={\left(H^{T}C_{\omega T}^{-1}H+Y^{T}C_{\omega T}^{-1}Y\right)}^{-1}$$


### 3.2. GDOP of TDOA and AOA Hybrid Positioning

The definition of GDOP requires FIM to be reversible, that is, det(J) ≠ 0. Usually analyze the necessary and sufficient conditions of det(J) = 0, and then use the equivalence principle of the proposition and its inverse proposition to obtain the corresponding condition of FIM reversibility.

CRLB of the hybrid positioning system has been obtained as shown in Equation (29), according to the definition of GDOP, GDOP is the square root of the trace of CRLB [22,23], which is the inverse of FIM [22,23].
(30)$$GDOP\left(s_{t}\right)=\sqrt{tr\left(CRLB\right)}$$
where tr(·) is the matrix trace.

## 4. Mayfly Optimization Algorithm (MOA) Station Deployment for TDOA/AOA Hybrid Positioning

### 4.1. Mayfly Optimization Algorithm

The Mayfly optimization algorithm is an improvement of the particle swarm algorithm. It combines the advantages of PSO (particle swarm optimization) [32,33], GA (genetic algorithm) [34] and FA (firefly algorithm) [35] to offer a powerful hybrid algorithm structure. Based on the social behavior of mayfly, crossover technology and local search are used. Assuming that the mayfly will always be an adult after hatching, the strongest mayfly can survive. The position of each mayfly in the search space represents a potential solution. The working principle of the algorithm is as follows. Two groups of mayflies, male and female, are randomly produced in the problem space as a candidate solution, represented by a d-dimensional vector $x=\left(x_{1},x_{2},\cdots ,x_{d}\right)$, and its performance is evaluated on a predefined objective function f(x). A speed vector is defined as $v=\left(v_{1},v_{2},\cdots ,v_{d}\right)$ to represent the change in the position of the mayfly. The flight direction of each mayfly is a dynamic interaction between individual and social flight experience. In particular, every time the mayfly will adjust its trajectory to reach the personal best position, which is the best position reached by any mayfly in the group [30].

The implementation of the mayfly optimization algorithm mainly includes the following steps:
Initialize the female and male mayfly populations and set the speed parameters.

Assuming that $p_{i}^{t}$ is the current position of male mayfly *i* in the search space at time step *t*, $q_{i}^{t}$ is the current position of female mayfly *i* in the search space at time step *t*.

The speed of the male mayfly is:(31)$$v_{ij}^{t+1}=v_{ij}^{t}+a_{1}e^{-\beta r_{p}^{2}}\left(pbest_{ij}-p_{ij}^{t}\right)+a_{2}e^{-\beta r_{g}^{2}}\left(gbest_{j}-p_{ij}^{t}\right)$$

The process that the female mayfly attracts the male mayfly is modeled, and the formula for calculating the speed of female mayfly is:(32)$$v_{ij}^{t+1}=\{\begin{array}{c} v_{ij}^{t}+a_{2}e^{-\beta r_{mf}^{t}\left(p_{ij}^{t}-q_{ij}^{t}\right)},if\;f\left(q_{i}\right)>f\left(p_{i}\right)\\ v_{ij}^{t}+fl*r,\;if\;f\left(q_{i}\right)\leq f\left(p_{i}\right) \end{array}$$
where $v_{ij}^{t}$ is the speed of mayfly *i* (*I* = 1, 2, …, *n*) in dimension *j* (*j* = 1, 2, ..., *d*) at time step *t*, $\;p_{ij}^{t}\;$ and $q_{ij}^{t}\;$ is the position of male and female mayfly *i* in dimension *j* at time step *t*, *a*_1_ and *a*_2_ are positive attraction constant, *β* is a fixed visibility coefficient, *r_mf_* is the Cartesian distance between male and female mayfly. Finally, *fl* is a random walking coefficient, used when the female is not attracted by the male, so it flies randomly, and *r* is a random value in the range [−1 1].
2.Calculate the fitness value and sort to obtain *pbest_i_* and *gbest*.

*pbest_i_* is the optimal position that mayfly *i* has never been to. Taking into account the problem of minimization, the individual optimal position *pbest_i_* at the next time step *t* + 1 is calculated by the following formula:(33)$$pbest_{i}=\{\begin{array}{c} x_{i}^{t+1},if\;f\left(x_{i}^{t}\right)<f\left(pbest_{i}\right)\\ is\;kept\;the\;same,otherwise \end{array}$$
where *f* is the objective function, which characterizes the quality of the solution, and the global optimal solution, *gbest* is defined as:(34)$$\begin{array}{c} gbest\in \left\{pbest_{1},pbest_{2},\cdots ,pbest_{N}|f\left(cbest\right)\right\}\\ =\min\left\{f(pbest_{1}),f\left(pbest_{2}\right),\cdots ,f\left(pbest_{2}\right)\right\} \end{array}$$

The crossover operator represents the mating process of two mayflies. Parents are selected in the same way as females are attracted to males. In particular, the selection can be random or based on their fitness function. Subsequently, the best female matches the best male, the second best female matches the second best male, and so on. The result of the cross is two offspring, the results are as follows:(35)$$\begin{array}{c} offsping1=L*male+\left(1-L\right)*female\\ offsping2=L*female+\left(1-L\right)*male \end{array}$$

Male refers to the male parent, and female refers to the female parent. *L* is a random number within a specific range, and the initial velocity of the offspring is set to zero.
3.Update the positions of male mayflies and female mayflies in turn, and mate.

The next position can be obtained by adding a velocity $v_{i}^{t+1}$, it can be expressed as:(36)$$p_{i}^{t+1}=p_{i}^{t}+v_{i}^{t+1}$$
(37)$$q_{i}^{t+1}=q_{i}^{t}+v_{i}^{t+1}$$
where $p_{i}^{0}\cup \left(p_{min},p_{max}\right)$, $q_{i}^{0}\cup \left(q_{min},q_{max}\right)$
4.Calculate the fitness and update *pbest* and *gbest*;5.Whether the stop condition is met, if it is met, exit and output the result, otherwise repeat Steps 3–5.

The mayfly optimization algorithm can be described as the pseudo code shown in Algorithm 1. In the pseudo code, the update process of the female and male mayfly populations are combined.
**Algorithm 1: Mayfly optimization algorithm.****Input**← objective function f(x), male population size (M1), female population size (M2), visibility coefficient (β), initial velocity for male mayfly ($v_{m}$), initial velocity for female mayfly ($v_{n}$), maximum iteration (T), *a*_1_ and *a*_2_ are positive attraction factors, solution dimension (d), population size (L)**Output**→ Optimal Solution (*gbest*)
1: **begin**
2: Evaluate all solutions according to the objective function f(x)
3: Find the best value from all solutions (*gbest*)
4: **while** t ≤ T
5:    find *pbest* by (33), the best solution of each male mayfly 
6:    **for** I = 1:M2
7:     **for** j = 1:d
8:          Adjust female velocity by (32)
9:          Adjust female positions by (37)
10:    **end for**
11:    **end for**
12:    **for** I = 1:M2
13:     **for** j = 1:d
14:          Adjust male velocity by (31)
15:          Adjust male positions by (36)
16:     **end for**
17:     Update *pbest*

18:    **end for**
19:    Rank male mayflies
20:    Rank female mayflies
21:    Mate the mayflies
22:    Evaluate the offspring
23:    Separate offspring to male and female randomly
24:    Replace the worst solution with the best new one
25:    Update *pbest* and *gbest*
26: **end while**
27: **end**

### 4.2. Flow Chart of MOA Station Deployment for Hybrid Positioning

This section mainly shows the process of station layout optimization. Firstly, the objective function must be determined. Here, the proposed objective function is defined as the average GDOP of the multi-base station TDOA/AOA hybrid positioning system. The dimensionality of the target variable is related to the base station number (*n*) and the dimensionality (D) of the base station coordinates (2D or 3D). The objective region is determined according to factors such as the communication range of the positioning system. Population initialization needs to determine the population size of female mayflies and male mayflies, initial position, initial velocity, visibility coefficient β, and positive attraction factors a1 and a2. The maximum number of iterations T must be determined, which is the parameter that determines the end of the optimization algorithm [36]. The specific optimization process is shown in Figure 2. Here we initialize the population size to 20, and the maximum number of iterations T = 100.

## 5. Simulation

In this section, some simulations are proposed to evaluate the positioning performance of 3D TDOA and AOA using the geometric distribution of base stations optimized by GDOP. The simulated positioning system is aimed at LTE mobile communications in a complex environment, the positioning base stations are carried by UAVs, and the semi-passive positioning of mobile terminals is achieved by acquiring the measured values of TDOA and AOA through the acquisition of random access signals.

### 5.1. Confirm the Model of Optimal Station Deployment

The optimization model of the multi-station semi-passive positioning system mainly includes three parts: independent variables, constraints, and objective functions.

#### 5.1.1. Independent Variables

To achieve the optimal deployment of the positioning system, it is actually to determine the location where the positioning base station is located to minimize the positioning error in the target area [37,38]. Therefore, the independent variable of the optimal deployment model about the semi-passive TDOA and AOA positioning system is the coordinate position of the receiving base station. Assuming that the number of base stations is *n* + 1, if the station is deployed in a three-dimensional space, the coordinates of the base station are $\left(x_{i},y_{i},z_{i}\right)$, where *i* = 0, 1, …, *n*. The self-variation of the optimal station deployment model in the three-dimensional space is $\left(x_{0},y_{0},z_{0},x_{1},y_{1},z_{1},\cdots ,x_{N},y_{N},z_{N},\right)$, it is 3(*n* + 1) dimensions in total.

#### 5.1.2. Constraint Conditions

When the positioning system optimizes the deployment of stations, it will have certain requirements on the base station deployment area and target positioning area according to actual conditions, such as geographic location, communication conditions, noise interference, target movement range, etc., and the positioning accuracy must be based on the location of the base station within a certain range and the determined target in a certain area. Assuming that the location of the base station is **s**, the location of the target radiation source is **x**, the location area of the positioning system is $R_{1}$, and the target area is $R_{2}$, then the constraint conditions of the optimal deployment model are $\left\{s\in R_{1},x\in R_{2}\right\}$.

#### 5.1.3. Objective Function

The fitness function value is an important evaluation index, which is a criterion for retaining and eliminating individuals. For multi-station semi-passive positioning systems, selecting the GDOP value that characterizes the positioning accuracy index is more conducive to the optimization of station deployment. The smaller the GDOP value, the higher the positioning accuracy. In this paper, the average GDOP value of the positioning system for target positioning in a certain area is selected as the fitness function [39]. It is expressed as:(38)$$Fitness=\frac{\sum_{m=1}^{M}GDOP_{m}}{M}$$
where M is the number of the targets, m=1,2,⋯M. Firstly, traverse the positioning area to obtain the coordinates of M targets, and then find the average of the GDOP values, it is the objective function. The MOA optimization algorithm is to find the layout coordinates of the base stations when the objective function is the minimum, which is also the parameter estimated by the MOA algorithm.

### 5.2. Simulation of Optimized Layout for the Unmanned Aerial Vehicle (UAV) Airborne Multi-Base Station

#### 5.2.1. Simulation Scenario 1

With reference to the geometric distribution of satellite positioning, assuming that the receiver clock error is zero, the geometric distribution of the positioned target and the satellite satisfies that the positioning error is small when the observation vector is perpendicular to each other, while the positioning error is large when the satellite is in the same direction corner. Therefore, the base station distribution range setting in the simulation surrounds the target range, and adjacent areas are perpendicular to each other [40].

The simulated range of the targets is defined as x = [−200, 200], y = [−200, 200], z = 0, and The distribution ranges of the UAV airborne multi base-stations are $x_{1}=\left[-200,-180\right]$, $y_{1}=\left[-200,-180\right],\;z_{1}=\left[1,5\right],\;x_{2}=\left[180,200\right],\;y_{2}=\left[-200,180\right],\;z_{2}=\left[1,5\right],\;x_{3}=\left[-180,200\right],\;y_{3}=\left[180,200\right],\;z_{3}=\left[1,5\right],$ $x_{4}=\left[-200,-180\right],\;y_{4}=\left[-180,200\right],\;z_{4}=\left[1,5\right]$. The error parameters in the positioning algorithm are as follows, standard deviation of TDOA measurement error is $\sigma_{d}=10m$, standard deviation of AOA measurement error is $\alpha_{1}=\alpha_{2}=0.01rad$, where $\alpha_{1}$ is about the azimuth, $\alpha_{2}$ is about the elevation angle. The above description is the parameters of simulation scenario 1. Multiple algorithms, such as MOA, PSO, ABC (artificial bee colony algorithm) [38], and GA, are used to optimize the layout of the base station, and the number of simulation parameters population size and the number of iterations are the same. The population size is 20, and the number of iterations is 100. The simulation results are shown in Figure 3 and Table 1. Compared with the other three algorithms, the Mayfly optimization algorithm achieves better optimization results and obtains the smallest GDOP average value, which means that the optimized station layout can achieve the best positioning accuracy. However, if there is a lack of time to run enough number of iterations, the PSO algorithm is better than MOA (for a number of iterations from about 1 to about 22); GA is better than MOA for a number of iterations from about 6 to about 16.

In the case of MOA optimized station deployment, the contour map of GDOP is shown in Figure 4. The positioning of the base station is isotropic, and the results of the station layout around the rectangular area are shown. The optimal distribution of the base stations is close to the edge of the positioning area and is an approximate diamond shape.

#### 5.2.2. Simulation Scenario 2

Simulation scenario 2 is an area of 400 × 400 m, the targets to be located are distributed in the middle area, and the base stations of the positioning system are distributed around. The simulated distribution range of the targets is defined as x = [−180,180], y = [−180,180], z = 0, and the base station distribution around the targets region. The distribution range x_1_ = [–200, 160], y_1_ = [–200, –160], z_1_ = [1, 3], x_2_ = [160, 200], y_2_ = [–200, 160], z_2_ = [1, 3], x_3_ = [–160, 200], y_3_ = [160, 200], z_3_ = [1, 3], x_4_ = [–200, –160], y_4_ = [–160, 200], z_4_ = [1, 3]. The base stations are all unfixed and the coordinates need to optimize calculation. The simulation is aimed at the complex ruin environment which is shown in Figure 5, the collapsed building has two or three floors. The collapse model includes stacked collapse, supported collapse and completely collapsed. And the measurement errors of each base station are different, which is brought about by the complexity of the channel environment. Assume that the standard deviations of TDOA distance measurement error and AOA angle measurement error in positioning are, respectively, $\sigma_{d1}=12m,{\;\sigma }_{d2}=10m,\sigma_{d3}=8m,$ $\alpha_{1\_1}=\alpha_{1\_2}=0.01rad$, $\alpha_{2\_1}=\alpha_{2\_2}=0.02rad$, $\alpha_{3\_1}=\alpha_{3\_2}=0.015rad$, $\alpha_{4\_1}=\alpha_{4\_2}=0.017rad$ where $\alpha_{i\_1}$ is about the azimuth, $\alpha_{i\_2}$ is about the elevation angle.

Without optimizing the layout, place the base stations arbitrarily within the distribution range of the base stations. The GDOP diagram is shown in the Figure 6. The positioning accuracy of the entire area is greater than 3 m.

The simulation deployment station is limited by the actual conditions of the ruin environment. There are three types of deployment station ranges, namely, the surrounding layout, the double-sided deployment station, single-sided deployment and the vertical adjacent deployment station. The simulation process is as follows: first, the MOA algorithm is used to optimize the deployment of the three ranges, and the simulation results are the optimal base station coordinates and the optimal fitness value (the minimum value of the average GDOP value). Then, solve the GDOP value under the optimal base station layout, and draw the contour line of GDOP. The simulation results of the three cases are shown in Table 2 and Figure 7, Figure 8, Figure 9 and Figure 10. Note that the red triangles in the figures represent the location of the base station.

From the simulation results of four base station layout optimization, it can be seen that for the positioning of targets in a fixed area, the deployment of stations has a significant impact on the positioning accuracy. Comparing Figure 6, Figure 7, Figure 8, Figure 9 and Figure 10, the GDOP value in most of the positioning areas in Figure 6, Figure 7 and Figure 8 is less than 3 m, and the positioning performance of the station placement results is better than the random station placement shown in Figure 5. Most of the GDOP values shown in Figure 10 are larger than 3 m, and the positioning accuracy is the worst, so deploying base stations on one side of the positioning area is the most unsuitable way to do so. The deployment with the highest positioning accuracy is to arrange stations around the target area, and the stations are close to the center of the area. The layout of the base stations with the worst positioning accuracy is to deploy the stations on single side of the target area.

#### 5.2.3. Positioning Simulation of Five Base Stations in Simulation Scenario 2

The five-base station positioning system improves the positioning accuracy by increasing the number of base stations when the deployment conditions are not good. The simulation is based on the vertical adjacent distribution of four base stations, adding a fixed base station. The location of the base station is divided into three situations. The main base station S_0_ is fixed and the coordinates are respectively (–200, 200, 3) or (180, –180, 3) or (–200, –200,3). The simulation results are shown in Table 3 and Figure 11, Figure 12 and Figure 13.

From the simulation results of the five base stations, it can be seen that when the deployment of the base stations is restricted, the purpose of improving the positioning accuracy can be achieved by increasing the number of base stations. The location of the added base station will also affect the positioning accuracy. The fitness values of the three different main base station fixed points are: 2.4308, 2.2625, and 2.3165, which all exceed the fitness value of the four base station distribution. The average GDOP under the above seven layout situations is shown by a histogram as shown in Figure 14.

In addition, we have also carried out the optimization problem of adding more base stations with fixed base stations in the four corners of the area, with 6–8 base stations we respectively obtained the best fitness equals 2.1314, 7, 2.0208, 1.8803. It can be seen that the accuracy of positioning is improved when the number of the base station is increased. However, these simulation results ignore the problems of positioning settlement, ranging and angle measurement error, and cost caused by the addition of base stations. If we want to reach accurate conclusions, this will be carried out in the follow-up research work.
moa01: Four stations surrounding the target region;moa02: Four stations are distributed on both sides of the target area;moa03: Four stations are distributed on the vertical side of the target area;moa04: Four stations are distributed on single side of the target area;moa05: Five base stations add station S_0_= (−200, 200, 3) to the distribution of moa03;moa06: Five base stations add station S_0_= (180, −180, 3) to the distribution of moa03;moa07: Five base stations add station S_0_= (−200, −200, 3) to the distribution of moa03.

## 6. Discussion

This paper proposes an optimal geometric configuration algorithm for UAV airborne base stations, it is in TDOA and AOA hybrid positioning in the specified area by using the Mayfly optimization algorithm. The average GDOP value of the target in the positioning area is used as the objective function to optimize the deployment of the base stations. The simulation and analysis of the optimization of the different number of base stations, compared with other station layout methods, such as PSO, GA and ABC. MOA is less likely to fall into local optimal, and the error of regional target positioning is surely reduced. By simulating the deployment of four base stations and five base stations in various situations, MOA can achieve a better deployment effect. The dynamic station configuration capability of the multi-station semi-passive positioning system has been improved with the UAV. The simulation verifies that the MOA algorithm has better optimization performance than other swarm intelligent algorithms. Using the same population size and the same number of iterations, the MOA optimization result can obtain the smallest fitness value. In the case of the restricted area of the ruined environment, combined with the complexity of the channel, the measurement errors of TDOA and AOA in different directions are different. The optimized layout of the four base stations and the five base stations was simulated, and the optimal layout of the different stations was obtained. According to the comparison of the station layout scheme, the positioning system layout should avoid the same side as much as possible. Increasing the number of base stations can significantly increase the positioning accuracy. In a word, the positioning accuracy can be improved by optimizing the geometric distribution station.

## Figures and Tables

**Figure 1 sensors-21-07484-f001:**
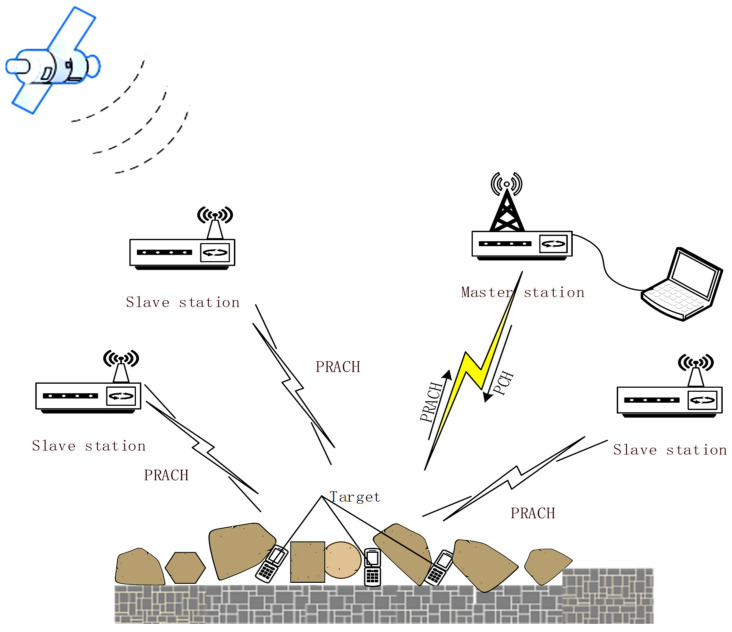
Semi-passive positioning system in ruin environment composed of 1 master base station and *n* slave base stations.

**Figure 2 sensors-21-07484-f002:**
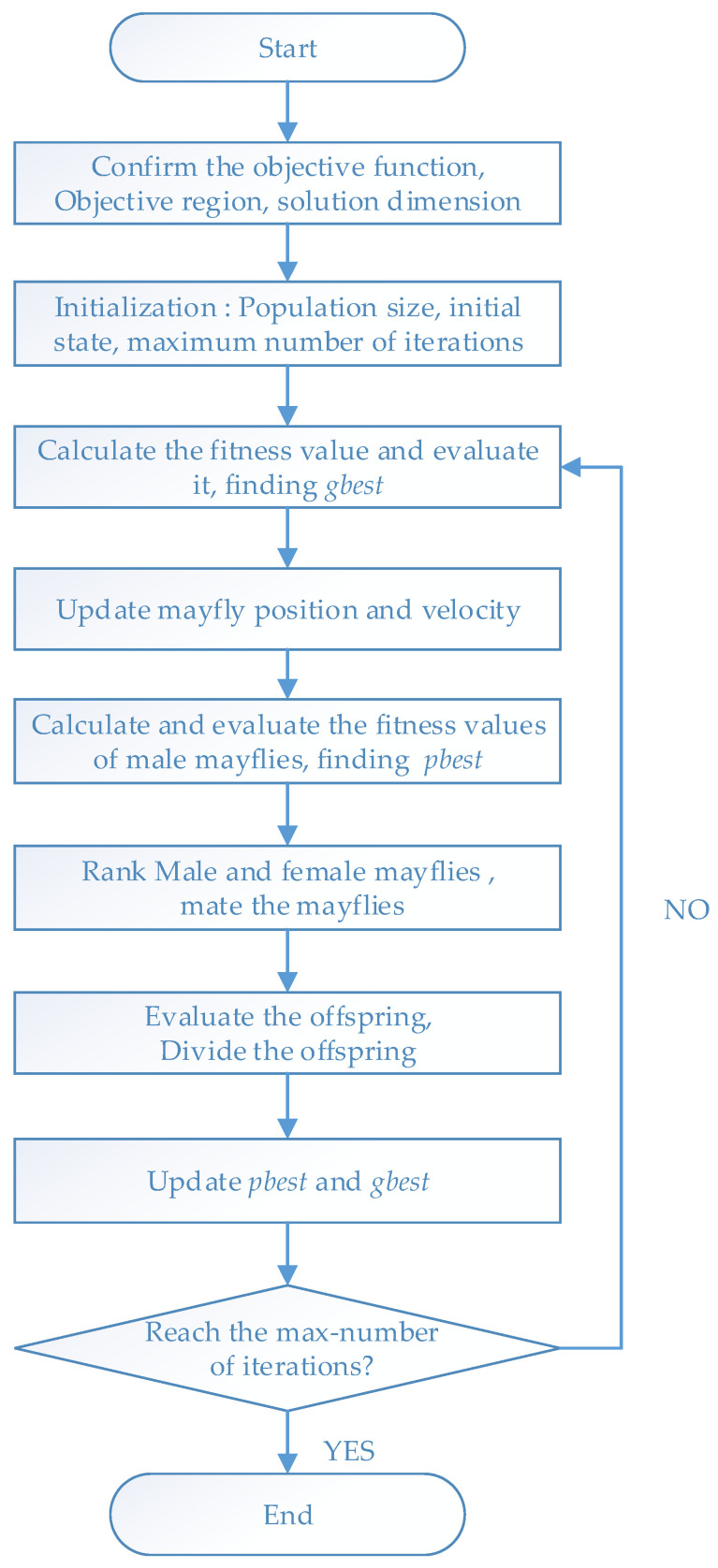
Flow chart of MOA station deployment for hybrid positioning.

**Figure 3 sensors-21-07484-f003:**
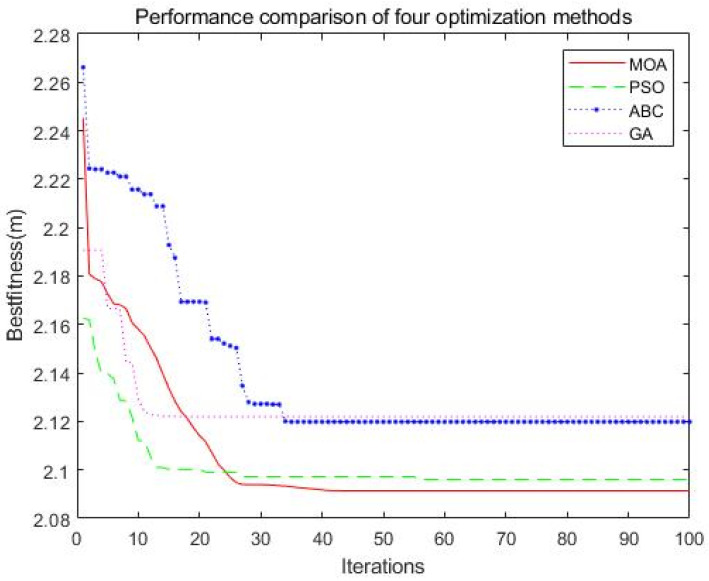
Iterative convergence process of four optimization algorithms for fitness function.

**Figure 4 sensors-21-07484-f004:**
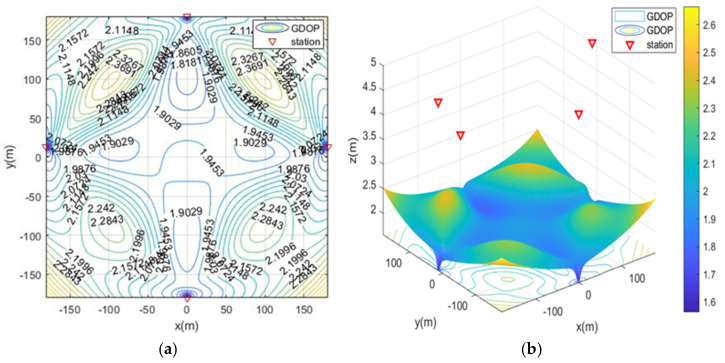
The geometric dilution of precision (GDOP) of MOA optimized station deployment in scenario 1. (**a**) 2D contour map; (**b**) 3D contour map.

**Figure 5 sensors-21-07484-f005:**
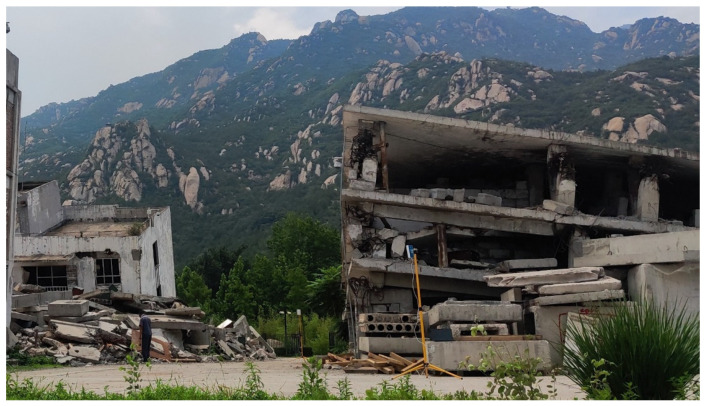
The ruin environment with different standard deviations of the measurement error in different directions.

**Figure 6 sensors-21-07484-f006:**
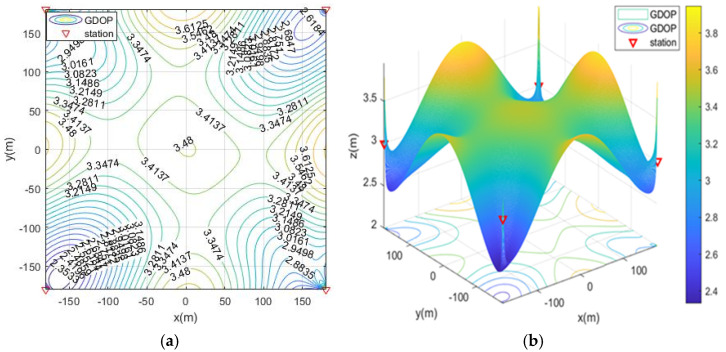
GDOP contour map without optimized station deployment. (**a**) 2D contour map; (**b**) 3D contour map.

**Figure 7 sensors-21-07484-f007:**
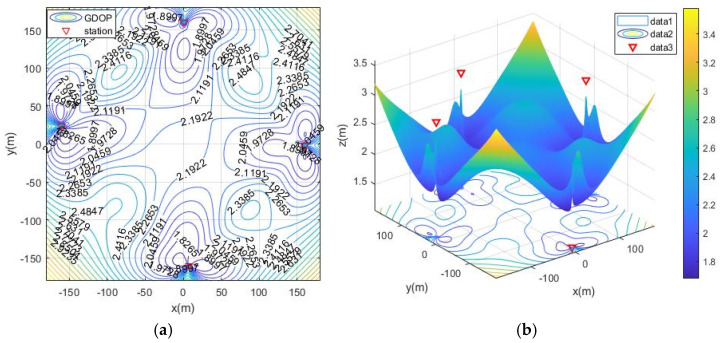
GDOP contour map of optimization results with 4 base stations located surrounding the positioning area. (**a**) 2D contour map of moa01; (**b**) 3D contour map of moa01.

**Figure 8 sensors-21-07484-f008:**
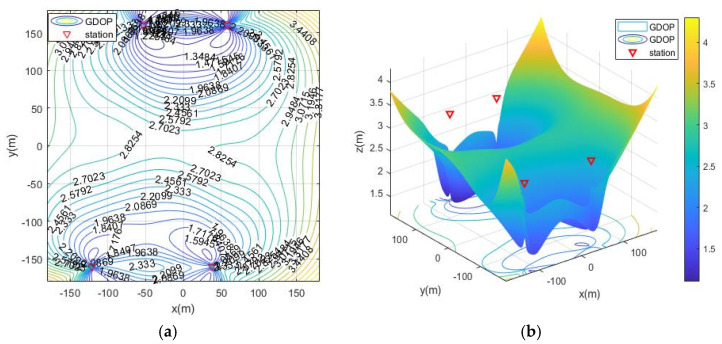
GDOP contour map of optimization results with 4 base stations located on both sides of the positioning area. (**a**) 2D contour map of moa02; (**b**) 3D contour map of moa02.

**Figure 9 sensors-21-07484-f009:**
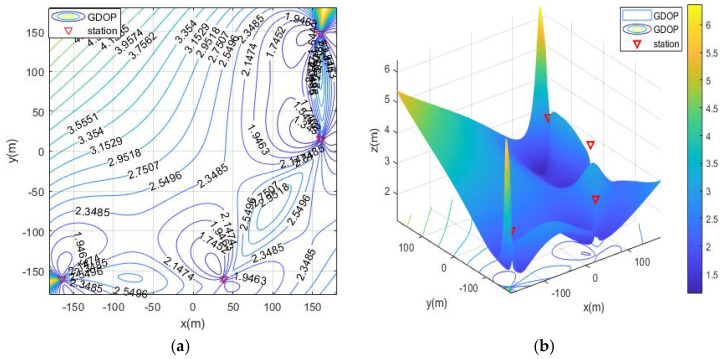
GDOP contour map of optimization results with 4 base stations located on vertical adjacent sides of the positioning area. (**a**) 2D contour map of moa03; (**b**) 3D contour map of moa03.

**Figure 10 sensors-21-07484-f010:**
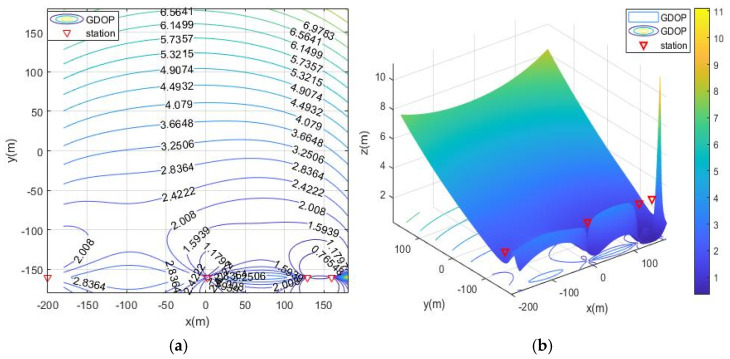
GDOP contour map of optimization results with 4 base stations located on single-side of the positioning area (**a**) 2D contour map of moa04; (**b**) 3D contour map of moa04.

**Figure 11 sensors-21-07484-f011:**
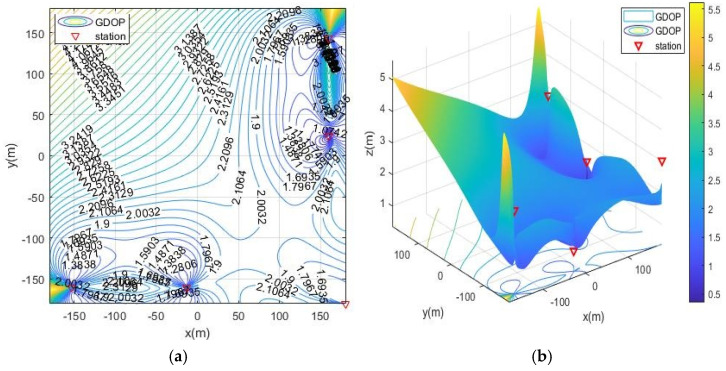
GDOP map of five base stations optimization results with a fixed station (–180, 180, 3). (**a**) 2D contour map of moa05; (**b**) 3D contour map of moa05.

**Figure 12 sensors-21-07484-f012:**
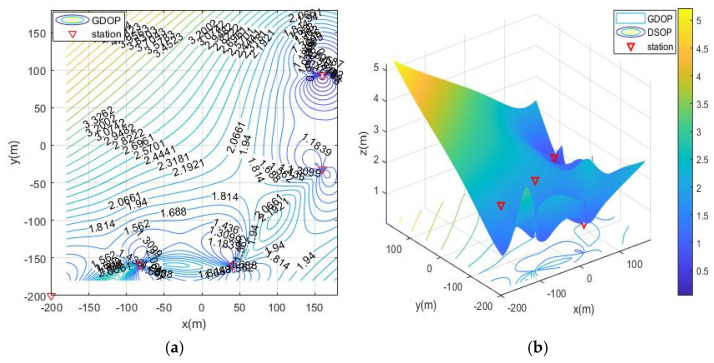
GDOP map of five base stations optimization results with a fixed station (–200, –200, 3) (**a**) 2D contour map of moa06; (**b**) 3D contour map of moa06.

**Figure 13 sensors-21-07484-f013:**
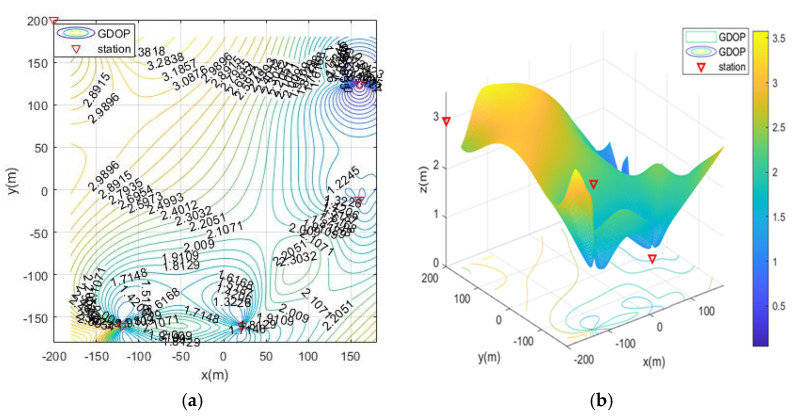
GDOP map of five base stations optimization results with a fixed station (−200, 200, 3) (**a**) 2D contour map of moa07; (**b**) 3D contour map of moa07.

**Figure 14 sensors-21-07484-f014:**
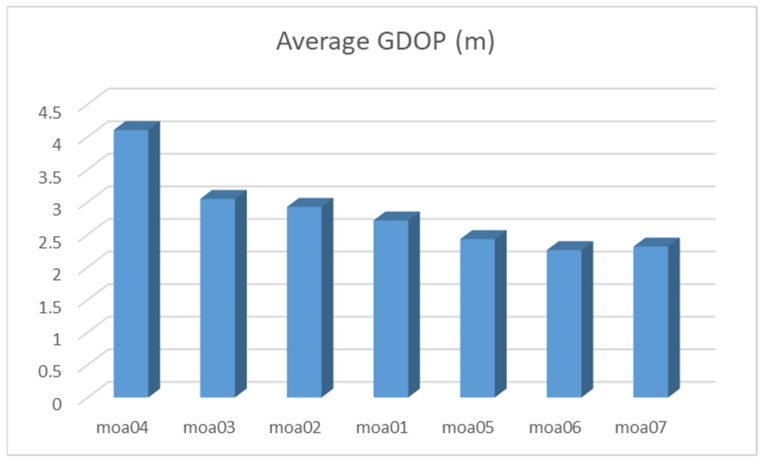
The average GDOP under the above seven base station layout situations.

**Table 1 sensors-21-07484-t001:** Optimized results of the four algorithms under scenario 1 ^1^.

Algorithm	Best Fitness (m)	Best Solution					Convergence
			S_0_ (m)	S_1_ (m)	S_2_ (m)	S_3_ (m)	
GA	2.1174	x	−11.5593	180.4777	27.4115	−180.3046	22
		y	−186.2205	6.2025	181.0768	6.1282	
		z	4.7982	4.8517	3.0579	2.7393	
PSO	2.096	x	2.1101	180	−10.2916	−180	56
		y	−180	6.4877	180	7.4060	
		z	3.9209	3.1175	2.1361	4.0180	
ABC	2.1198	x	1.7987	180	−3.9197	−181.4895	35
		y	−180	13.9337	190.0083	22.5678	
		z	5	5	5	5	
MOA	2.0914	x	−8.9245 × 10^−6^	180.0001	−1.7622 × 10^−6^	−180	42
		y	−180	13.3577	180	13.3577	
		z	5	5	5	5	

^1^ S_0_ is the master station. S_1_, S_2_ and S_3_ are the slave stations.

**Table 2 sensors-21-07484-t002:** Optimized results of the four stations deployments under scenario 2 ^1^.

Deployment	Best Fitness (m)	Best Solution				
			S_0_ (m)	S_1_ (m)	S_2_ (m)	S_3_ (m)
Surrounding (moa01)	2.7173	x_1_	7.5222	160,	0.9865	−160
		y_1_	−160	−2.2527	160	23.6182
		z_1_	1.0603	3	2.9971	3
Double-side (moa02)	2.9254	x_2_	37.7234	−121.025	58.2313	−53.0542
		y_2_	−160	−160	160	160
		z_2_	3	3	3	3
Vertical Adjacent (moa03)	3.0493	x_3_	−164.5013	38.5807	160	160
		y_3_	−160	−160	14.5338	147.1652
		z_3_	3	3	3	3
Single-side (moa04)	4.104	x_4_	−200	2.1497	128.7016	160
		y_4_	−160	−160	−160	−160
		z_4_	3	3	3	3

^1^ S_0_ is the master station. S_1_, S_2_ and S_3_ are the slave stations.

**Table 3 sensors-21-07484-t003:** Optimized results of the five station deployments.

S_0_	Best Fitness (m)	Best Solution				
			S_1_ (m)	S_2_ (m)	S_3_ (m)	S_4_ (m)
(180, −180, 3) moa05	2.4308	x	−152.9574	−12.8670,	160	160
		y	−160	−160	143.4871	25.0102
		z	3	1	3	1.7007
(−200, −200, 3) moa06	2.2625	x	−82.6562	40.8520	160	160
		y	−160	−160	−31.9998	93.7736
		z	3	1	1	1
(−200, 200, 3) moa07	2.3165	x	−120.2850	21.7966	160	160
		y	−160	−160	−11.6254	124.7243
		z	1	1	1	1

## Data Availability

Not applicable.

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
