# Peer review of "An Optimal Geometry Configuration Algorithm of Hybrid Semi-Passive Location System Based on Mayfly Optimization Algorithm"

_sensors, 2021, doi:10.3390/s21227484_

Round 1

Reviewer 1 Report

The authors presented a proposal of geometry configuration algorithm of elements of location system based on MOA for the location of mobile phones (for search people who have a mobile phone with them) in a complex environment. Subject is very interesting and important but very complex. Such systems are very much needed for practical applications in the case of natural disasters, terrorist acts and for military applications.

I. Following are the detail comments:

1. The presented system is not entirely passive, but rather it should be called semi-active (or semi-passive), because the searched phones are induced to emission (are asked) by the master base station which is one the elements of the location system.

2. The relationship between MOA elements (parameters estimated in MOA) and information generated by base stations and positions of the localized objects (mobile phones) are not clearly shown in the manuscript.

3. It would be worth taking into account the structure of the debris depending on whether small houses or high-rise buildings have been destroyed.

4. The simulation results are very interesting and show the potential usefulness of the developed solution for the mobile phones location system. However, its wide application will require many verification tests in real-word conditions.

5. Line 340 – a comment to Figure 3: when is a lack of time to run enough number of iterations, the better than MOA is PSO (Particle Swarm Optimization) algorithm (for a number of iterations from about 1 to about 22); Genetic Algorithm (GA) is better than MOA for a number of iterations from about 6 to about 16.

II. Some editing suggestions (in my opinion). It is proposed to review the entire text in this respect:

1. Line 82 – the solution concerning CRLB and GDOP are described in Section 3 of manuscript not in Section 2.

2. Line 83 – should be Section 4 not Section 3.

3. Line 85 – simulation results are presented in Section 5 not in 4.

4. Line 87 – conclusions (discussion) are presented in Section 6 not in 5.

5. Line 108, 109 – dominant station S0, in line 94/95 it was called master station MS (unless they are different objects).

6. Line 285 – word "confirm" should be written with a capital letter.

7. Line 342 – Table 1: It is not explained what is: S1, S2, and S3. One can only guess this.

8. Line 342 – Table 1 – lack of units of variables x, y, z, and Best fitness parameter, whereas in Table 2 (Line 381) units of variables x, y, z, and Best fitness parameter are given.

Reviewer 2 Report

Overall, the manuscript needs significant improvement of grammar.  There is prolific use of incomplete sentences.

In-text citations need to be fixed to include et al. instead of etc. 

Title should not have an abbreviation (MOA) in it.   Similarly, the abstract uses numerous abbreviations without defining them; please define them in the text prior to using them.

The literature review in the Introduction is difficult to follow and the overall flow could be greatly improved.  Please also clarify if anyone else has even considered applying these methods to the debris environment.  The literature review discuses different algorithms but doesn’t discuss how these have been specifically applied.   Also highlight how this paper is addressing a gap in the literature.

The paper discusses the target (versus the slave/master stations).  Where is the target on Figure 1?  Please label.

Carefully review the variables that are included in the paper.  What is c and delta*ti in Equation 1?  What is e in equation 7?  Where do x, y, and z come from in equations 8, 9, etc.?  What is Cq in equation 21?  What is cbest in Equation 34?

Is the (1,:) necessary in AT,i sub-equation in equation 7?

Define || - || in equations 2 and 3.

Section 3.2 is unclear; please clarify this section.

Line 207, it is mentioned that the advantages of other algorithms are leveraged through MOA.  What other algorithms?  Please be specific.

Line 273, it states that T must be determined; how is T determined?  Please clarify.

Equation 38 – remove the double t in “Fittness”.

Line 328, more clearly indicate that the i = 1 to 4 for the z coordinate.

Figure 3 – include units on the y-label.

Be consistent in the paper with your abbreviation of the mayfly optimization algorithm – is it MOA or MA?  See figure 3, table 3, figure 14.

Table 1.  Provide units on the Best Fitness.

In table 1, PSO and MOA are close in GDOP but result in vastly different positions.  Does this matter?  Please discuss.

Line 371 – 374, it is unclear why you are choosing these layouts.  Please include references from state of the art that provide support for these choices.  It is unclear how these configurations are implemented or what these configurations mean.  Please clarify. 

Include legends on Figure 7 – 9, match what is shown on Figure 6.

Figure 14, include y label units.

Based on your brief analysis, the GDOP is decreased with 5 base stations.  What if you had 6 or 7 or more base stations?  Will it continue to decrease?  Is there a trade off for more base stations?  When will adding more base stations not be beneficial.  Please develop this further.
